# The Expression of Prolactin Receptors in Benign Breast Tumors Is Not Associated with Serum Prolactin Level

**DOI:** 10.3390/jcm10245866

**Published:** 2021-12-14

**Authors:** Olena Kolomiiets, Oleksandr Yazykov, Artem Piddubnyi, Mykola Lyndin, Ivan Lukavenko, Volodymyr Andryushchenko, Anatolii Romaniuk, Roman Moskalenko

**Affiliations:** 1Department of Pathology, Sumy State University, 40022 Sumy, Ukraine; o.kolomiets@med.sumdu.edu.ua (O.K.); a.piddubny@med.sumdu.edu.ua (A.P.); n.lyndin@med.sumdu.edu.ua (M.L.); 2Private Clinic “MRIYA”, 40004 Sumy, Ukraine; o.yazikov@med.sumdu.edu.ua (O.Y.); i.lukavenko@med.sumdu.edu.ua (I.L.); v.andryshenko@med.sumdu.edu.ua (V.A.); 3Department of Surgery, Sumy State University, 40007 Sumy, Ukraine; 4Department of Medical Biochemistry and Biophysics, Umeå University, 90736 Umeå, Sweden; 5Ukrainian-Swedish Research Center SUMEYA, Sumy State University, 40022 Sumy, Ukraine

**Keywords:** prolactin, prolactin receptors, benign breast tumors

## Abstract

The role of prolactin (PRL) and its receptors in the initiation and development of benign breast tumors (BBT) has not been sufficiently studied. An imbalance in the system of hormone homeostasis is crucial in the development of BBT. In particular, an association between elevated prolactin levels and the development of BBT has been reported. Our study showed no significant differences between PRL receptor (PRL-R) expression in BBT tissue under normal and elevated serum PRL levels. There was also no significant correlation between age, PRL-R expression in BBT tissue, intact tissue, and PRL level in the serum. There was a strong significant correlation (*p* < 0.01; r = 0.92) between PRL-R expression in BBT samples and intact breast tissue, which did not depend on the serum PRL level. There was also no significant difference in the expression of the proliferative marker Ki-67 in BBT tissues from women with normal and elevated levels of serum PRL (*p* > 0.05). No signs of PRL and its receptors were detected in the BBT cystic fluid women with elevated serum PRL levels. In summary, our prospective study showed that the expression of PRL-R in the tissue of BBT and physiological breast tissue does not depend on the level of serum PRL.

## 1. Introduction

The development of the breast is related to the regulation of hormones such as estrogen, progesterone, prolactin, and growth factors. An imbalance in the system of hormone homeostasis is crucial in the development of its pathology [1,2]. Recently, scientists have focused on the role of the prolactin receptors (PRL-R) and serum prolactin (PRL) in the carcinogenesis, prognosis, and metastasis of breast cancer (BC) [3].

It was shown that secretion of prolactin also occurs outside the pituitary gland, namely by the endometrium, lymphocytes, decidual membrane, breast, and prostate. In these organs, PRL has cytokine-like properties, binds to PRL-R on the cell surface and cytoplasm, and activates the signaling pathways Jak2/Stat5 and Map Kinase [4,5]. 

There are physiological and pathological reasons for increased PRL levels in blood. Physiological causes include pregnancy, childbirth, and the postpartum period. Pathological causes of increased PRL are adenomas, inflammatory conditions, and pituitary surgery [6]. Hereditary idiopathic hyperprolactinaemia, which is not associated with pituitary adenoma, has also been reported [7]. 

A. Alyson et al. shown an important role of PRL-R-induced global gene expression in cell proliferation [8]. A. Sutherland et al. found that a normal level of PRL in the serum is associated with the risk of BC and an increased incidence of metastasis [9]. Some patients with BC had elevated PRL levels and PRL-R expression, which might be associated with resistance to therapy and a poor prognosis [9]. 

S. J Howell has shown that exogenous PRL causes the proliferation, migration, and invasion of BC cell lines in vitro and that the majority of BC cells express PRL-R. PRL also increases the clonogenicity of primary human BC cells in soft agar [10]. Excess PRL has also been shown to increase the frequency and rate of tumor growth in many rodent models of spontaneous and carcinogenically induced breast tumors [10].

S. Gill et al. showed no correlation between PRL-R expression and tumor grade score, size, or axillary lymph node status, suggesting that the assessment of PRL-R would not be useful as a prognostic factor [11].

C. Manhès et al. showed that local over-expression of PRL in developing mammary glands induces dramatic functional and morphological lesions, but that this is not related to carcinogenesis [12]. However, M. Nicol et al. concluded that the great majority of patients with operable breast lesions have normal serum levels of PRL. The elevation of serum PRL levels had the same incidence in both groups of patients: with benign and malignant breast tumors [13]. 

However, the latest studies produced controversial results. K.A. O’Leary et al. demonstrated that PRL can modulate the incidence and phenotype of breast tumors and by so doing elevates the risk of breast cancer [14]. On other hand, I.Y. Hachim et al. showed that PRL has anti-tumorigenic properties in HER-2 positive BC, through the induction of ALDH^hi^ stem-like cell differentiation, and limits their stem-like aggressive properties [15].

This prospective study was aimed at detecting the relationship between the serum PRL level and tissue expression of PRL-R in a consecutive series of patients with BBT. 

## 2. Materials and Methods

### 2.1. The Ethics Committee

The study was approved by the ethics committee of the Medical Institute of Sumy State University (protocol No 3/02, 9 February 2021).

### 2.2. Research Design

The study used the biological material (serum, tumor tissue and adjacent intact tissue of breast, and cystic fluid) from 16 female patients with BBT. The mean age of the women was 27.9 ± 1.55 years, the age ranged from 19 to 39 years. Surgical interventions and blood sampling were performed at the private clinic ‘MRIYA’ after gaining the patient’s informed consent.

### 2.3. Evaluation of Serum PRL Level 

Blood samples for the PRL tests were collected in the morning, at least two hours after the patient had woken, to reduce the sleep-induced peak level of PRL. No restrictions on food, fluid, and physical activity were required. The patient had relaxed for at least 30 min before the sample collection [16]. Venous blood was collected from each patient, centrifuged, and stored in a freezer. Serum samples were ECLIA tested for PRL with a Cobas immunoassay analyzer system. The threshold level of serum PRL was considered to be 444.78 mU/L (or 21.18 ng/mL) [16].

Based on the level of serum PRL, 8 patients (group I) with high (more than 444.78 mU/L) and 8 patients (group II) with low (less than 444.78 mU/L) were selected for the study. During the study, both adjacent (subgroup A) and BBT (subgroup B, contain fibroadenomas (FA) and fibrous cystic disease (FCD)) tissue samples from each person were studied (Figure 1). 

At the same time, 6 fluid samples from cysts taken from patients of group II were examined to detect the presence of PRL and PRL-R (taken in cases of cystic changes). Cystic fluid was collected using ultrasound-guided fine-needle aspiration, centrifuged (13,000 rpm for 5 min), and the supernatant was stored in a freezer.

### 2.4. Histology and Immunohistochemistry

Breast tissue was fixed for 24 h in a 10% solution of neutral buffered formaldehyde, dehydrated, and embedded in paraffin. For immunohistochemistry (IHC), sections with a thickness of 4 μm were subjected to standard deparaffinization and dehydration in xylene and inversed ethanol gradient. After antigen retrieval in 0.1 M citrate buffer (pH 6.0) at a temperature of 97–98 °C, indirect IHC was performed:

I stage: incubation with primary antibodies for 30 min (t = 37 °C);

II stage: incubation with secondary antibodies (UltraVision ONE HRP Polymer, Thermo Scientific, Waltham, MA, USA) for10 min (t = 37 °C). Diaminobenzidine (Thermo Scientific, Waltham, MA, USA)—5 min (t = 37 °C). Nuclei were counterstained with Mayer’s hematoxylin. 

For this study, we used the following antibodies: anti-prolactin receptor mAb (clone B6.2, dilution 1:200, Thermo Scientific, Waltham, MA, USA) and anti-Ki-67 mAb (SP6, dilution 1:600, Thermo Scientific, Waltham, MA, USA). All photos were captured with a digital visualization system based on a Zeiss Primo Star microscope with a Zeiss Axiocam ERc 5-s digital camera and software package ‘Zen 2.0’ (Carl Zeiss, Jena, Germany). Generally, immunohistochemical staining of tissue was performed according to the method previously described in our works [17]. At least 6 different fields of view (FOV) were analyzed for each sample. The IHC results were presented as a mean number of PRL-R- and Ki-67-positive cells per FOV with a diameter of 1 mm.

### 2.5. SDS-PAGE and Western-Blot 

The total protein concentration was measured with a ULTROSPEC 2100 Pro UV-visible spectrophotometer (Biochrom US, Holliston, MA, USA) at the wavelength of 280 nm. 

All proteins were separated by SDS-PAGE (Tris-HCl 7.5% gels). Cyst content was mixed with 2% β-mercaptoethanol containing 2x SDS loading buffer and pre-heated to 95 °C for 5 min. Each well was loaded with an equal protein amount: 20 µg of protein per well. SDS-PAGE gels were stained with a Coomassie brilliant blue R-250 stain. 

Proteins were blotted to PVDF membrane (Bio-Rad, Hercules, CA, USA) and blocked with 5% skimmed milk solution. Separate membranes were probed with anti-human prolactin mAb (clone PRL02, dilution 1:1000, Invitrogen, Waltham, MA, USA) and anti-prolactin receptor mAb (clone B6.2, dilution 1:1000, Thermo Scientific, USA) primary antibodies, followed by goat-anti-mouse HRP-conjugated secondary antibodies (dilution 1:2000, Thermo Scientific, Waltham, MA, USA). Membranes were developed with Pierce ECL Western Blotting Substrate (Thermo Scientific, Waltham, MA, USA). All captures were made with a Kodak Digital Image station 2000R (Eastman Kodak Company, Rochester, NY, USA). Each experiment was repeated in two replicates.

### 2.6. Statistical Analysis

The normality of data distribution was checked using the Shapiro–Wilk test. Student’s *t*-test was applied for analysis of data with a normal distribution. Mann–Whitney’s U-test was applied for nonparametric data sets. The results were considered statistically significant with a probability of more than 95% (*p* < 0.05). Pearson correlation was used to correlate age, PRL level, and PRL-R expression in tumor and adjacent tissue. Statistical analysis was performed in Microsoft Office Excel 2016 with the add-on AtteStat (version 12.0.5). All graphs were made with GraphPad Prism 8.0.

## 3. Results

### 3.1. Macroscopic Study of Breast Tissue

Macroscopically, FA had the form of a rounded node surrounded by a capsule, with clear edges, dense consistency, grayish-white color, and sized from 0.5 to 3.5 cm (Figure 2). FCD in section had a whitish color with the presence of fibrous bands that intertwined with each other and with adipose tissue. There were cysts of different sizes and number, their surface was smooth and had a gray-pink color.

### 3.2. Histology and Immunohistochemistry

The adjacent tissue had a dense (fibrous) consistency and loose connective tissue around lobes (Figure 3A, general background). The epithelium of the lobe had signs of functional activity; in the lumen of individual glands and ducts eosinophilic secretion was revealed. Interlobular and intralobular connective tissues had no significant differences. Ducts had different sizes and uneven lumens. Duct epithelium had a mature appearance and showed signs of secretion (Figure 3A, insert).

PRL-R was expressed with a mixed membrane-cytoplasmic pattern in the glandular and ductal epithelium (Figure 3B). The level of expression of the proliferation marker Ki-67 was moderate, it was found in some epithelial cells of glands and single fibroblasts (Figure 3C).

FA tissue (Figure 3D) consisted of stromal and glandular tissue. The stromal component was represented by fibrous tissue, with a reduced number of cells. Glandular structures were partially compressed by connective tissue, had an enlarged lumen, and were lined by monomorphic cubic and prismatic epithelium (Figure 3D, insert). FCD regions had a change in the ratio between the glandular and stromal component, as well as sclerosis and cystic changes in the ducts, with areas of simple ductal hyperplasia.

IHC of BBT tissue with anti-PRL-R antibodies revealed its moderate expression in some cells of the glandular epithelium within FA and FCD tissue (Figure 3E). The number of cells with a positive Ki-67 nuclear staining was nonsignificant, with prevalence in epithelial cells of BBT glands (Figure 3F).

The adjacent breast tissue of patients from group II had an increased amount of connective tissue around lobes. The glandular epithelium had dark nuclei and a rich cytoplasm, and there were signs of edema and hyperaemia (Figure 4A). This tissue had a moderate expression of PRL-R in glandular epithelium (Figure 4B). The number of Ki-67-positive nuclei in the epithelial cells of the glands was low (Figure 4C).

Microscopically, FCD was manifested by the stromal fibrosis and a reduction of the number of glands and focal lymphocytic infiltration, which was localized near cysts and dilated ducts (Figure 4D). Cysts were also found, lined with cubic epithelium with a weak light cytoplasm and rounded nuclei. The contents of the cysts showed eosinophilic fine-grained or homogeneous secretion.

The IHC of PRL-R revealed its moderate expression in the glandular epithelium and connective tissue (Figure 4E). The marker Ki-67 was mainly expressed in the nuclei of epithelial cells of glands and ducts. Some areas of increased proliferative activity of the tissue were detected (Figure 4F).

### 3.3. Statistical Analysis

The mean number of PRL-R-positive cells per FOV in group II (39.97 ± 2.1) was higher than in group I (35.56 ± 3.6), but no significant difference between both groups was detected (*p* > 0.05) (Figure 5A).

We also detected no significant difference in Ki-67 expression in the BBT tissues of women with normal and elevated serum PRL levels; group I samples had 16.6 ± 4.63 cells with positively stained nuclei per FOV, in group II—17.96 ± 2.84 positive cells (*p* > 0.05) (Figure 5B). 

Pearson’s correlation analysis did not reveal a significant correlation between age, PRL-R expression in BBT, adjacent tissue, and serum PRL level of patients of both groups (Figure 6). There was a moderate nonsignificant association (r = 0.31) between the expression of PRL-R in the tissue of BBT and adjacent breast tissue of patients, as well as a weak relationship (*p* = 0.29) between the expression of PRL in adjacent breast tissue and the serum PRL level (for all—*p* > 0.05).

Correlation analysis in group I revealed a strong positive correlation between the expression of PRL-R in the tissue of BBT and the level of PRL in patients with a non-elevated level (*p* < 0.05; r = 0.8).

In addition, there was a strong positive nonsignificant correlation between the age of patient and the expression of PRL-R in the tissue of BBT (*p* > 0.05, r = 0.64), and a moderate positive nonsignificant correlation between age and the expression of PRL-R in adjacent breast tissue and serum PRL (*p* > 0.05; r = 0.37 and r = 0.33, respectively). In the group of patients with normal PRL levels, there was a nonsignificant weak positive correlation between PRL-R levels in adjacent tissues and serum PRL levels.

An interesting result was revealed in the correlation analysis of data from group II. A strong significant correlation (*p* < 0.01; r = 0.92) was found between PRL-R expression in BBT tissue and adjacent breast tissue. Analysis of other values of this group did not show strong reliable correlations, but there was a tendency towards a negative relationship of moderate strength between age and PRL-R expression in BBT tissue and adjacent tissue (*p* = −0.31 and r = −0.43). There was also a weak positive relationship between PRL-R expression in breast tissues (tumor and adjacent tissue) and serum PRL levels. 

### 3.4. SDS-PAGE and Western-Blot

The cyst liquid contained 3.5–12.7 mg/mL of proteins. With SDS-PAGE a similar protein composition was found in the majority of samples. Only one sample consisted predominantly of proteins, with a molecular weight of approximately 55 kDa. Based on that, this sample was excluded from further experiments. 

With Western-blot, we did not reveal the presence of any proteins of interest (PRL-R and PRL) in all samples (Figure 7).

## 4. Discussion

Recent studies do not provide a clear picture of PRL’s role and its receptors in the differentiation, growth, and development of breast tumors. [18,19]. There have been a number of studies on the association of PRL and its receptors with benign and malignant breast tumors [19,20]. On the other hand, I. Hachim et al. found that the expression of PRL-R is reduced in invasive BC (21.4%), compared with BBT (80%) and carcinoma in situ (60%) (*p* = 0.003498) [21]. There was also no association between the gene polymorphism of PRL-R (PRLRI146L and PRLRI176V variants) with breast tumor growth and proliferation [22].

Despite the significant amount of information about the role of PRL in the development of breast oncopathology, our data did not show a correlation between PRL level and the expression of its receptors with the development of BBT.

In this study, we compared the expression of PRL-R and proliferation marker Ki-67 in BBT and adjacent tissue from women with normal and elevated serum PRL levels. To further verify the presence and possible role of PRL and PRL-R in the development of BBT, the cyst fluid of BBT from a group of patients with elevated serum PRL levels was examined by SDS-PAGE and Western blot.

We found no significant difference between the expression of PRL-R in samples of BBT from patients with normal and elevated levels of serum PRL (*p* > 0.05). The correlation analysis of group II revealed a strong significant correlation (*p* < 0.01; r = 0.92) between PRL-R expression in BBT tissue and adjacent breast tissue. This result is quite natural because, in patients with elevated serum PRL, the number of receptors also increases, not only in tumors, but also in adjacent healthy tissues. This shows the absence of an exclusive effect of PRL on the tissues of BBT in comparison to adjacent tissues.

The association of PRL with breast epithelial proliferation has been proven beyond doubt [23]. However, the comparison of Ki-67 expression in the samples of groups I and II in our study did not show a significant difference between them (*p* > 0.05) (Figure 6). Therefore, elevated serum PRL levels do not lead to an increased epithelial proliferation in BBT tissue.

On Western-blot, we did not detect the presence of PRL and PRL-R in cystic fluid from the BBT of group II (with elevated serum PRL) (Figure 7). It seems that an increased level of PRL in the serum does not lead to its automatic increase in body fluids. On the other hand, the correlation analysis of the results of group I revealed a strong positive significant relationship between the expression of PRL-R in the tissue of BBT and the normal level of PRL (*p* < 0.05; r = 0.8).

## 5. Conclusions

Our study shows no significant difference between PRL-R expression in BBT tissue under normal and elevated serum PRL levels. There is also no significant correlation between age, PRL-R expression in BBT and adjacent tissue, and serum PRL level. In patients of both groups, there was a strong significant correlation (*p* < 0.01; r = 0.92) between PRL-R expression in BBT tissue and adjacent breast tissue. There was also no significant difference in the expression level of the proliferative marker Ki-67 in tissues of BBT from women with different levels of serum PRL (*p* > 0.05). No signs of PRL and its receptors were detected in the cystic fluid of BBT from women with elevated serum PRL levels.

In summary, our prospective study shows that the expression of PRL-R in BBT and physiological breast tissue does not correlate with the level of serum PRL. 

In the future, it is planned to study the effect of PRL-R expression on the development, prognosis, and metastasis of BC.

## Figures and Tables

**Figure 1 jcm-10-05866-f001:**
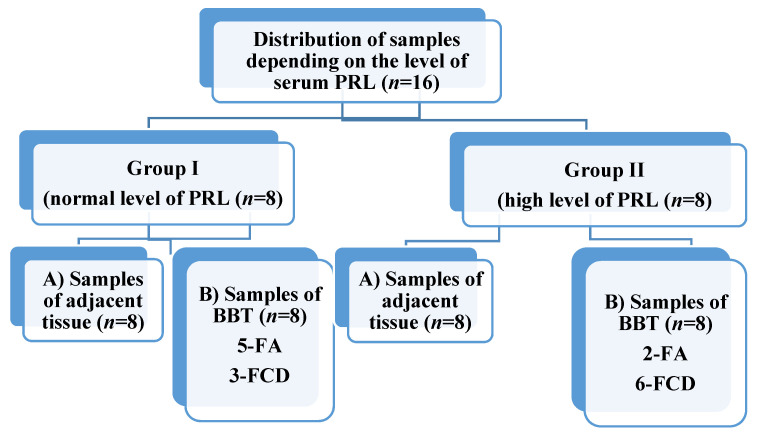
Distribution of breast tissue samples. PRL—prolactin, FA—fibroadenoma, FCD—fibrous cystic disease.

**Figure 2 jcm-10-05866-f002:**
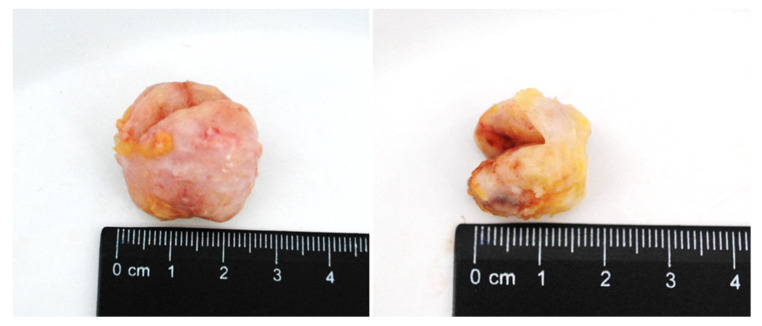
BBT (fibroadenoma) of the breast in section.

**Figure 3 jcm-10-05866-f003:**
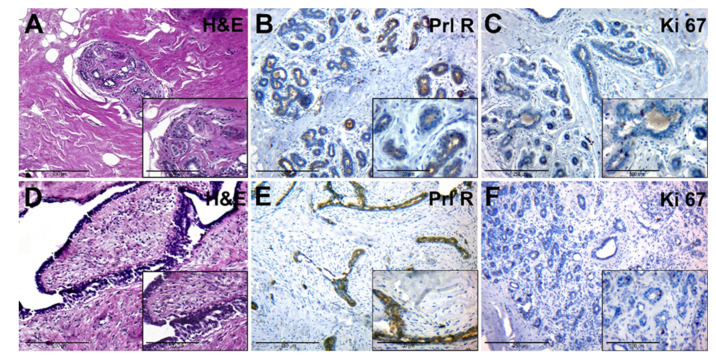
Histological and immunohistochemical examination of tissue samples from patients with normal levels of serum PRL. (**A**–**C**)—adjacent breast tissue of patients with normal levels of serum PRL (group IA). (**A**)—hematoxylin-eosin staining; (**B**)—IHC detection of PRL-R expression; (**C**)—IHC expression of Ki-67. (**D**–**F**)—BBT tissue (FA) of patients with normal levels of serum PRL (group IB). (**D**)—hematoxylin and staining; (**E**)—IHC detection of PRL-R expression; (**F**)—IHC expression of Ki-67. Magnification (including inserts) is indicated in the lower-left corner of the image as a marker.

**Figure 4 jcm-10-05866-f004:**
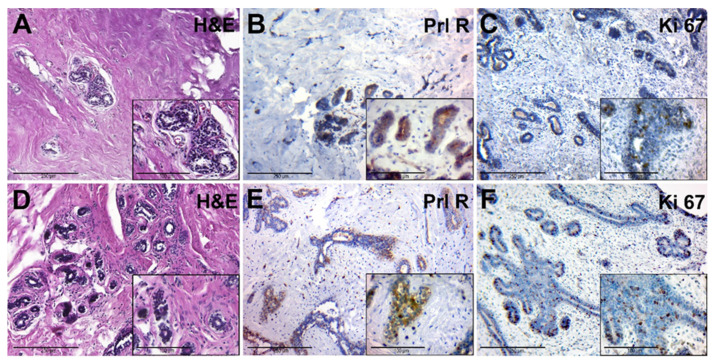
Histological and immunohistochemical examination of tissue samples from patients with elevated levels of serum PRL. (**A**–**C**)—adjacent breast tissue of patients with elevated levels of serum PRL (group IIA). (**A**)—hematoxylin-eosin staining; (**B**)—IHC detection of PRL-R expression; (**C**)—IHC expression of Ki-67. (**D**–**F**)—BBT tissue of patients with elevated levels of serum PRL (group IIB). (**D**)—hematoxylin and staining (group 2B); (**E**)—IHC detection of PRL-R expression; (**F**)—IHC expression of Ki-67. Magnification (including inserts) is indicated in the lower left corner of the image as a marker.

**Figure 5 jcm-10-05866-f005:**
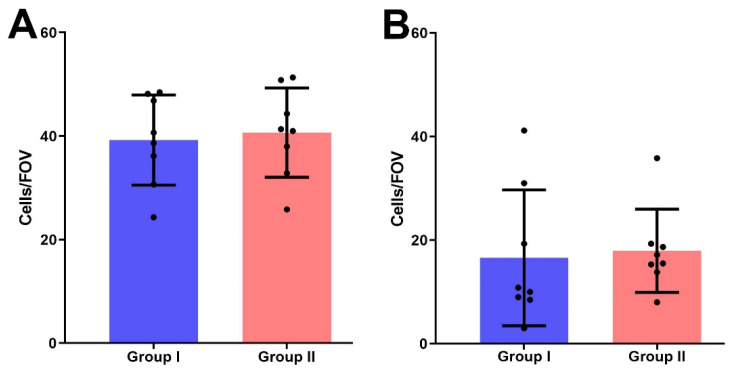
The number of IHC-positive cells for the expression of PRL-R (**A**) and Ki-67 (**B**) in BBT tissues of I and II groups per field of view.

**Figure 6 jcm-10-05866-f006:**
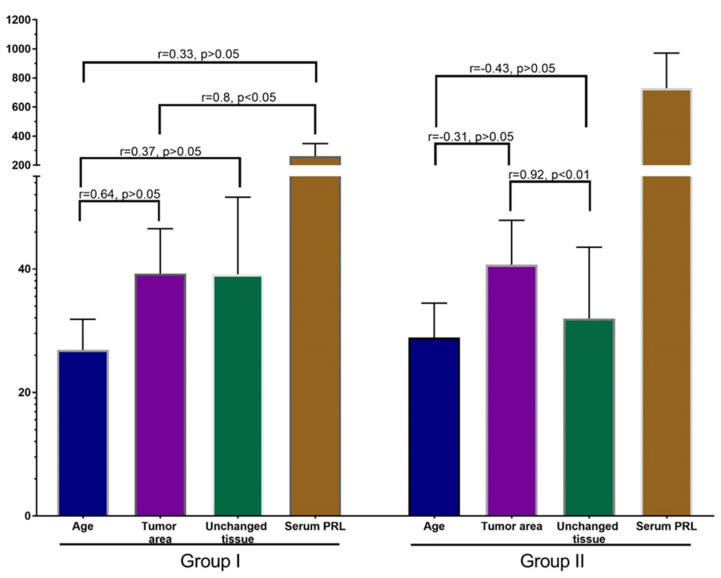
Correlation analysis of patients’ age, PRL-R expression in BBT and adjacent tissue, and serum PRL. Values: age—years, PRL-R—IHC-positive cells per FOV, Serum PRL—mU/L.

**Figure 7 jcm-10-05866-f007:**
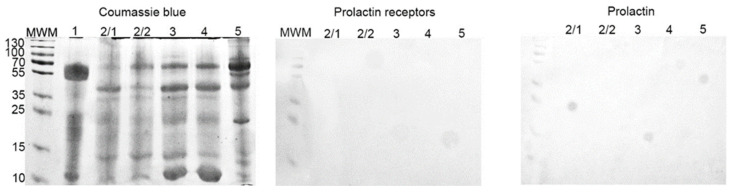
SDS-PAGE and Western-blot of cyst liquid from group IIB. SDS-PAGE gel stained with Coomassie brilliant blue; Western-blot with anti-PRL-R and anti-human PRL mAbs, respectively.

## Data Availability

Data available within the article.

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
