# Peer review of "The Expression of Prolactin Receptors in Benign Breast Tumors Is Not Associated with Serum Prolactin Level"

_jcm, 2021, doi:10.3390/jcm10245866_

Round 1
Reviewer 1 Report
This article aims at correlating prolactin levels on the development of benign breast tumors. The manuscript has several flaws which the authors should take into consideration that currently preclude the publication in this journal. In addition, a thorough review of the grammar/spelling of the manuscript is required. Importantly, abbreviations must be defined the first time they appear and also be defined in figure legends. Also, attention must be given to concentration units.
Abstract:
Besides giving insights of the adopted methodology and describing the key findings of the study, the abstract should clearly identify the problem the authors addressed and why it is important/relevant, as well as appropriate conclusions based on their findings and also broader implications/impact of the study. The current abstract must be rewritten accordingly.
Introduction:
The introduction included in the manuscript failed in defining the purpose of the work and its significance, including the specific hypothesis being tested. Despite some background on the role of prolactin in breast development and how it can become pathological, the authors do not describe previous reports on the role of prolactin in benign breast pathology, which apparently have already been published, as they say it has not been sufficiently studied (lines 59-61).
Materials and Methods:
The authors used 16 patients as the study population. Is this a reasonable number to stablish any realistic correlations with statistical significance and confidence?
Given the importance of statistical analyses for this type of studies, why was GraphPad Prism not used for those analyses instead of MS Excel?
How much protein mass was used to load SDS-PAGE gels?
Results:
Overall, this section must be revised and subsections renamed and well-organized.
If the role of prolactin is better understood on pathophysiological development, samples of breast cancer patients should be used as controls.
Importantly, WB experiments must be repeated. As they are, it seems that the samples collected did not contain prolactin, which is the “biomarker” of interest in the present study. Saying that “Protein traces indicated possible degradation of proteins” (line 241) is not acceptable.
Discussion/Conclusion:
Overall, this section must be revised and really focused on the result interpretation having in perspective previous studies. An overall and integrative conclusion must be provided. Moreover, future perspectives should be addressed.
Author Response
Dear reviewer!
Thank you for your efforts in reviewing this paper. We highly appreciate your comments and believe that they will improve our article.
We have changed and extended the abstract. We focused on our main results and put them in the proper place in the abstract.
We checked all abbreviations and assumed that they had a proper place and definition at the first appearance time.
We have included the purpose of our study and information on the study of prolactin in the context of its effect on benign breast pathology in the section "Introduction."
We share the reviewer's concern about the small number of samples in this study. In general, this study is part of a larger project to study the effect of prolactin on carcinogenesis involving samples of malignant tumors. The project is currently under development, but we believe these results could be published and have some scientific value.
In our opinion, MS Excell with addon Attestat is more convenient for us. Nevertheless, we used GraphPad Prism to visualize graphs since it's more understandable and "reader-friendly." We will consider the use of GraphPad Prism as the main statistical tool.
Each well was loaded with an equal protein amount - 20 ug.
As we noted above, this paper is part of a bigger project about breast carcinogenesis. The main goal of our study was to estimate the correlation of serum prolactin level with the expression of prolactin receptors. Since we were not focused on the analysis of cancer tissue, we did not include such samples in the study. The adjusted physiological breast tissue was used as a control.
We ran WB experiments in replicas and got identical results with the complete absence of prolactin and prolactin receptor signals. Based on that, we assume that probes did not contain these proteins. We agree with the reviewer that the statement about protein degradation is incorrect and deleted it from the manuscript.
Also, we have corrected the spelling of prolactin concentration units (corresponds to as in source 12).
We revised the section "Discussion/Conclusions" and added an integrative conclusion. Also, we added the statement about future perspectives of research.

Reviewer 2 Report
The manuscript is original because it explores the field of benign breast tumours (BBT) considering the PRL and PRL-R (which is quite new) proving mainly the non-correlation between BBT and serum level of PRL and presence of PRL-R in the breast tissue. It is written appropriately and is correctly designed, although the population examined is numerically low. From the practical point of view it does not provide any contribution, but give a substantial contribution in the research field where most publications are focused on the role of PRL in the appearance and development of breast cancer and not on BBT.
I have some suggestions for authors. I think the title is inconsistent with the argumentation reported in the study: from the title, the reader expects that prolactin and its cell receptors may play a role in the diagnosis of BBT. This I misleading because the authors are not proposing new histological or serological tests in diagnostics of BBT.
On page 4, lines 136 and 137, you mentioned for the first time two abbreviations without the extensive writing of the terms (FA and FCD). Please, make the due corrections.
In the conclusion, you should remark that unlike what has been shown by other similar studies in the field of breast cancer, in your preliminary study (considering the low number of participants), serum prolactin and its cell-receptors do not show any correlation with the presence or absence of BBT.
Author Response
Dear reviewer! We truly appreciate your valuable comments! Our study focuses on the role of prolactin in the development of benign breast pathology. The title of the article reflects the part of the overall project that this study is. We agree with you that there is some inconsistency between the title of the article and the results obtained, but given the planning of the overall project, we would prefer to keep the title as it is. We carefully checked all abbreviations and corrected the manuscript according to your comments. Thank you for the suggestion about the "Conclusion" section. We modified the text and made it more clear in accordance with your recommendations.

Round 2
Reviewer 1 Report
I would like to thank the authors for their responses. Please find below the comments and the points that should be addressed.
“Dear reviewer!
Thank you for your efforts in reviewing this paper. We highly appreciate your comments and believe that they will improve our article.
We have changed and extended the abstract. We focused on our main results and put them in the proper place in the abstract.”
1) Once again, I would like to thank the authors for their revision of the manuscript. Abstract section was indeed improved but proper introduction to the study is still missing. Why is it important to study serum level of PRL and presence of PRL-R in the breast tissue on the development of BBT? The purpose of the study should be clear. Accordingly, the title of the manuscript should also be reviewed to clearly correlate with the results presented.
“We checked all abbreviations and assumed that they had a proper place and definition at the first appearance time.”
2) The authors have thoroughly addressed this point throughout the manuscript. However, abbreviations should also be defined in the figure legends.
“We have included the purpose of our study and information on the study of prolactin in the context of its effect on benign breast pathology in the section "Introduction."”
3) Introduction has been revised and improved.
“We share the reviewer's concern about the small number of samples in this study. In general, this study is part of a larger project to study the effect of prolactin on carcinogenesis involving samples of malignant tumors. The project is currently under development, but we believe these results could be published and have some scientific value.”
4) The reader will not be aware that this study is part of a bigger project on breast carcinogenesis. Therefore, authors should mention that this study must be seen as a preliminary study due to the small number of patients studied. This would be of note in the conclusion section, for instance.
“In our opinion, MS Excell with addon Attestat is more convenient for us. Nevertheless, we used GraphPad Prism to visualize graphs since it's more understandable and "reader-friendly." We will consider the use of GraphPad Prism as the main statistical tool.”
5) No further suggestions on the software adopted. Concerning the graphs, in figure 6, label is missing in Y axis.
“Each well was loaded with an equal protein amount - 20 ug.”
6) This information must be part of the material and methods section.
“As we noted above, this paper is part of a bigger project about breast carcinogenesis. The main goal of our study was to estimate the correlation of serum prolactin level with the expression of prolactin receptors. Since we were not focused on the analysis of cancer tissue, we did not include such samples in the study. The adjusted physiological breast tissue was used as a control.”
7) This was just a suggestion to have a positive control on the study given that PRL and PRL-R are found on BC patient serum and tissue, respectively (according to the authors in the Introduction section).
“We ran WB experiments in replicas and got identical results with the complete absence of prolactin and prolactin receptor signals. Based on that, we assume that probes did not contain these proteins. We agree with the reviewer that the statement about protein degradation is incorrect and deleted it from the manuscript.”
8) The number of replicates should be mentioned on Materials and Methods section. Given the reproducible lack of detection of the proteins of interest (PRL and PRL-R), authors should consider in the future the use of a positive control for these proteins to validate the WB results.
“Also, we have corrected the spelling of prolactin concentration units (corresponds to as in source 12).”
9) Concentration units have been revised as previously suggested.
“We revised the section "Discussion/Conclusions" and added an integrative conclusion. Also, we added the statement about future perspectives of research.”
10) Integrative conclusion has been added to the Conclusion section helping on the take-home message even that the study was based on a small population. Authors should revise the future perspectives taking into consideration the main goal, the results and the conclusion of this study.
Author Response
Response to Reviewer 1 Comments ROUND 2
- Once again, I would like to thank the authors for their revision of the manuscript. The abstract section was indeed improved, but a proper introduction to the study is still missing. Why is it important to study serum level of PRL and presence of PRL-R in the breast tissue on the development of BBT? The purpose of the study should be clear. Accordingly, the title of the manuscript should also be reviewed to clearly correlate with the results presented.
Response: We changed the name of the manuscript following the results.
Insert into the abstract: “In particular, an association between elevated prolactin levels and the development of BBT has been reported.”
- The authors have thoroughly addressed this point throughout the manuscript. However, abbreviations should also be defined in the figure legends.
Response: Done!
- Concerning the graphs, in figure 6, label is missing in Y axis.
Response: This graph is only for visual comparison and demonstration of correlations and is not for comparing different values. We provide a detailed description of values (i.e., Y labels) for every bar in the figure legend.
- “Each well was loaded with an equal protein amount - 20 μg.” This information must be part of the material and methods section.
Response: Done!
- The integrative conclusion has been added to the Conclusion section helping with the take-home message even that the study was based on a small population. Authors should revise the future perspectives taking into consideration the main goal, the results and the conclusion of this study.
Response: “In summary, our prospective study shows that the expression of PRL-R in BBT and physiological breast tissue do not correlate with the level of serum PRL level in patients with normal and elevated levels of this hormone.
In the future, it is planned to study the effect of PRL-R expression on the development, prognosis, and metastasis of BC.”
We believe that there is no contradiction between the results of the expression of prolactin receptors in BBT and BC. This is because PRL may not be involved in the initiation of carcinogenesis but impact the prognosis and course of the tumor.
